# Beyond Software Development: Continuous Integration with coding agents

## Abstract

Coding agents are popular aids for software development today. Starting from code completion mechanisms in GitHub co-pilot, they have evolved much beyond programming to be active aids for software development by managing and maintaining a code-base via issue resolution. Resolving software issues takes care of program improvement tasks such as bug fixes and feature additions. Yet they do not contribute to the integration and operationalization of such changes into a complex software project. In this work, we study technical challenges that maintainers will face with integrating AI-generated suggestions such as build-problems and testing software systems. We compare how a variety of existing software engineering agents can cope with such operationalization and systems integration. This includes the open-source agent `OpenHands`, an existing agent to help in project builds called `ExecutionAgent`, and `USEAgent`, a general purpose SE agent ensemble. Furthermore, we introduce `USEAgentPlus`, an enhanced agentic system that extends prior agents by incorporating dedicated tools for environment management. This enables effective solutions for systems-integration tasks and facilitates experimental comparison. Our results suggest that the proposed solution outperforms existing approaches, achieving success on diverse open-source projects evaluated against a standard software engineering research benchmark. At a broad level, our work contributes in taking the automation offered by AI agents to the next stage of the software lifecycle - from software development and maintenance to software systems integration.

## 1 Introduction

Large language models (LLMs) profoundly reshaped the way developers write code to tackle tasks ranging from everyday automation to sophisticated applications when they are augmented with other tool chains (Yuan et al., 2024; Wang et al., 2024a; Jimenez et al., 2024). Automating coding tasks and facilitating software production, as such, have been one of the most active fields in AI research driven by the growing demand for autonomous tools that can accelerate development, reduce human burdensome. Asynchronous Software Development Agents show great promise to enhance developer workflows - working autonomously on issues (Zhang et al., 2024), reports (SonarSource), and generic instructions (Fruntke & Krinke, 2025) requesting little to even no human feedback until presenting a result. Research has made rapid progress towards producing quality patches (Jimenez et al., 2024; Yang et al., 2025) or automating quality tasks (SonarSource). What's next? State-of-the-art asynchronous tools start from an issue report and formulate a pull request (Nguyen & Nadi, 2022; Yetiştiren et al., 2023), a suggested change to the existing codebase, which is the current point to hand over to the maintainers, visualized in figure 1.

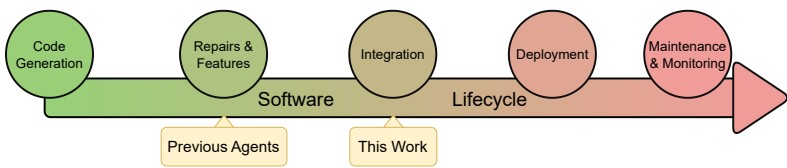

Figure 1: Overview: Progress of agentic systems for the software lifecycle

However, software production and evolution are much more than editing code in source files; maintainers must align requirements, user needs, and project vision (Linåker et al., 2024; Raman et al., 2020) with the code base. This is currently a blind spot in existing research of software engineering agentic systems, that largely focuses on the functional quality of patches or removing code defects. Even if existing tools can produce technically perfect code changes, software integration consists of more than patching (Hejderup & Gousios, 2022), and it is important to consider more aspects and spare maintainers from the increasing *automated pull request fatigue* (He et al., 2023; Kula, 2025). For a successful integration, we see three major obstacles in current agentic systems:

1. Poor retrieval of test- and quality-artifacts (see (Bouzenia & Pradel, 2025b)). Agentic systems must become better at managing execution environments and running tests, and not rely on additional external checks (e.g. re-iterating after an existing CI check (Maipradit et al., 2023), instead perform the checks before submitting a result). This effectively implies that agents must incorporate CI capabilities.

2. The misalignment between configuration languages and LLM' generalization capacity. For instance, the generated scripts often lack specificity and clarity for automation (Ghaleb & Rathnayake, 2025). This includes using wrong parameters or unnecessary elements in the output.

3. Finally, maintainers must make an educated decision on which changes to merge (Dias et al., 2021). Although this is largely subjective, maintainers can (and should) be supported in this decision. Agentic systems should automatically comprehend the runtime execution output after these changes and keep interacting with the environment until a user defined goal is achieved.

We argue that these key technical challenges to overcome these obstacles lies in **environment management** — a combination of the LLMs knowledge frame and awareness of its actions so far. Previous works (Bouzenia & Pradel, 2025b; Applis et al., 2025) outlines the difficulties for agentic systems in correctly installing dependencies and managing (virtual) environments, executing commands in wrong order or locations, and not recognizing the impact of commands that change environments. The complexity gets amplified once an agentic system must pay attention to more than one environment and program state, e.g. when facing a program fork that must be reconciled.

This motivated us to improve the capabilities of agentic systems by implementing a *evaluation step* that generates environment information for the *consensus memory* of the agent and provides a set of tools for version control systems and dependency management. These additions are made to the `USEAgent` (Applis et al., 2025), a framework-style software development agent that unifies testing, program repair and feature development, resulting in `USEAgentPlus`. That is also a major difference from existing work: Environment management is neither a standalone task (Bouzenia & Pradel, 2025b), nor a *given* to a SE agent (Yang et al., 2024), but instead forms an explicit focus point within other agentic development activities.

To evaluate the usefulness of our approach, we conducted extensive experiments. First, we present a study on the execution of 50 open-source projects written in seven programming languages to understand the effectiveness of `USEAgentPlus`. Moreover, we conduct a study on writing CI configuration scripts for popular Python libraries. We also present a study on repairing *SWE-Bench Verified* (Jimenez et al., 2024) to determine whether the changes lead to repair performance. Our experiment results indicate the proposed solution can execute 60% open sourced software projects by composing Bash scripts for a fresh environment, suggesting its high potential to work with human software engineers for testing their projects. This evidence also indicates that adding specialization to tasks beyond purely coding in the agentic ensemble is more fruitful (our work `USEAgentPlus`) than a generalist system (`OpenHands` CodeActAgent or `USEAgent`).

## 2 BACKGROUND

### 2.1 CONTINUOUS INTEGRATION

Continuous Integration (CI) refers to the practice in software production in which the project undergoes a controlled, continuous inflow of features and bug fixes to an evolving code base. This philosophy translates in practice to teams developing code against predetermined checks, such as compilation, code aduits and regression testing that are regularly executed, to ensure non-colliding

changes and a consistent quality standard. As software becomes larger and more complex, CI practice eases the complexity of its quality control, allowing for more efficient workflow by automating human efforts, as well as minimizing the delivery time.

Major software hubs like GitHub, GitLab or GitBuckets support CI through various means. The most prominent being *GitHub Actions* and *GitLab CI/CD* which allows users to define runtime environments through its infrastructure-as-code along with a set of directives that can be executed upon code change (new git commits) or repository event (new issue reports).

## 2.2 VERSION CONTROL SYSTEMS

Version control systems (VCS), such as git (Spinellis, 2012) are a major backbone of software development. Their 'diff'-style versioning for tracking code-changes from previous versions allows for roll-backs, feature selection and decentralized development (Blischak et al., 2016). To introduce decentralized changes, a merge-commit is made, reconciling differences into a joined version. Taking git-based development workflow as an example, CI testing runs automatically on each commit or pull request, identifying issues promptly. This practice mitigates "integration hell," where delayed merges lead to complex conflicts, by ensuring changes are validated in a controlled environment.

## 2.3 AGENTIC SYSTEMS FOR SOFTWARE ENGINEERING

ReAct-style agentic systems (Yao et al., 2023; Wang et al., 2024b) have seen rapid adoption in software engineering for two reasons: First, there are many common errors, such as non-compiling code, that can be eliminated with simple feedback. Most tools used by developers need little human interactions beyond a project-wide configurations (compilers, package managers, test-frameworks, etc.) and lend themselves towards agentic use through well documented command line interfaces. Second, a step-by-step approach towards successful solutions matches the workflow of developers. In most cases,, software development consists of an iterative change guided by knowledge and tests.

Both industry and research are making progress employing agents at all stages of the software lifecycle (Ruparelia, 2010), yet most mature tools are currently at the stage of repairing or enhancing programs by suggesting features (sketched in Fig.1). Integrating changes into the code-base, beyond a suggestion is the next step, which we aim to address in this work.

## 3 APPROACH

Agents struggle managing fragmented, yet connected information, such as software systems and their dependencies. Correct information can get lost in large contexts, and the agent can introduce changes to an environment (e.g. install a package). Another inherent difficulty is the *base*-environment that can vary, next to the versatility of build tools that different projects use. While it is difficult, we are in the fortunate position that it must be possible to build projects, and it is easy to verify the correctness of a result.

The issue of complexity is re-occurring through the progress of AI software development: From the starting points in code-completion, over code generation, to GitHub issue fix with agents, we face tasks that need more information and more steps for their correct solution. If environment-management is the technical barrier towards integration, are agentic systems capable?

**System Overview**  In this work, we investigate two existing LLM agentic system paradigms under this lens: (1) A pure *ReAct* Agent with software engineering specific instructions and low-level tooling (bash & file-editing). (2) The `USEAgent`-like multi-agent ensemble for software engineering with additional tooling (Applis et al., 2025). `USEAgent` is a modular framework that can be adapted for new tasks, wherein a *Meta* Agent orchestrates lower-level agents (e.g., code editing or test execution) and maintains *consensus memory* among them, distilled from the artifacts they generate. We propose `USEAgentPlus` by incorporating an environment probing stage, a self-critique mechanism for user task descriptions, and an re-iteration refinement strategy. From the foundation model perspective, our approach aligns with the intuition of Self-MoA-Seq (Li et al., 2025), that increasing in-model diversity can enhance the performance of agentic systems. From a software engineering perspective, enabling the agent to review historical tool calls and outputs reduces the

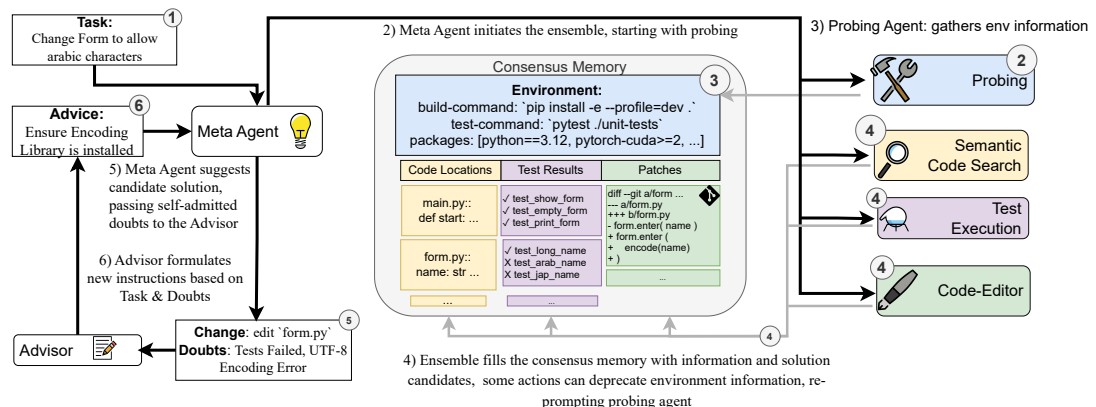

Figure 2: For a user task ①, the **probing agent** ② creates environment information, a pivotal point of the **consensus memory** ③. Once consensus is reached ④, the meta agent reports the result alongside self-admitted doubts ⑤ to the **advisor**, who acts as a judge and give out instructions for the next round of iteration ⑥.

hallucinations and lower the risk of the propagation of simple errors. The remainder of this section details our additional design contributions beyond USEAgent, and the overall workflow is depicted in Figure 2.

**Environment Probing Stage**    Agentic issues arising from dependencies are expected in practice: Projects are never run in a vacuum, and beyond the explicit dependencies written in the project configuration (e.g. the `pyproject.toml` for Python packages), there is a plethora of *hidden* dependencies at both project level and system level. To address existing shortcomings, we designed a `Probing Agent` that accumulates information about the system state beyond the project scope. It performs installation tasks and life checks on the project commands for exact content of an environment. The resulting environment knowledge is the pivotal point for the consensus memory; all memory, e.g. on test execution, is relative to one project state and one environment - if the project state changes, certain information (e.g. test status) gets automatically marked as obsolete and must be regenerated. Breaking changes, such as installs, deprecate more information including known project commands.

Since the system must first understand the current execution environment in order to provide accurate information about what actions are available, it should also be able to manage environments across different contexts, such as switching between folders. Thus in this step, the agent acquires knowledge centered on: identifying available testing tools, determining how test scopes can be constrained, verifying installed packages, build systems, and gathering operating system information.

**The Advisor Role**    The next design decision is introduce the `Advisor` agent, which re-iterates on output-candidates. In our workflow, the `Meta` agent is required to provide, alongside any requested action or patch, both an explanation of its underlying reasoning and a self-description of potential uncertainties. The `Advisor` subsequently reflects upon these doubts in relation to the consensus memory and the trajectory, and formulates a response that integrates: (i) a diagnostic assessment from a retrospective view; (ii) a prioritized sequence of corrective steps (e.g. quick fix list); (iii) relevant documentation, files, or folders for revisiting.

Our design is from a more fundamental perspective that an intelligent agentic system must first accumulate sufficient contextual knowledge from documents and environments rather than relying on external inputs (e.g. web search), to operate effectively, thereby approximating the workflow and reasoning processes of a human engineer (engineers may work on proprietary projects). An example of the agent's output is shown in Fig. 3. Previous work exploited *LLM-as-a-judge* specific to the produced artifacts (e.g. (Ruan et al., 2025)) or employ quality gates for retries (e.g. (Xia et al.,

2025)), which is insufficient in our context. Rewriting a better function or a code snippet might only rely on the context of the source file and natural language description but composing a bash script requires more diverse of knowledge of this project (e.g. programming languages or operating systems).

The historical view also allows our system to be scalable in multiple ways: Primarily, the number of iterations can be increased — we regard as a form of test-time scaling. Moreover, we can also instruct the advisor with task-adaptive instructions or introduce a human-in-the-loop (providing doubts and wanting clarifications). Effectively the `Advisor` helped us to overcome the issues observed with the latest frontier models: They ignore self-identified problems and refuse to address them, or abort tasks early.

**Consensus memory** In light of consensus memory proposed by Applis et al. (2025), which is a structured data object on the trajectories of the LLM components, we enhanced our agent architecture by maintaining a persistent shared memory accessible to all agents. Correctly executed actions from the probing agent and other agents are automatically written to this memory, ensuring that the current state and historical decisions remain visible and consistent throughout the system. Write operations to this shared memory are performed synchronously with the program flow rather than LLM agents actions to ensure consensus.

In Fig 2, we have presented a scenario in which the agents handle Python projects. After the probing agent sets up an environment, it summarizes and reports information on operating system (e.g. Linux distribution), file paths (e.g. the pytest folder) and packages (e.g. pytorch-cuda), that is written into the consensus memory, which is visible to other agents. This information can be vital for actions by the edit code agent. If further changes to the environment are required (e.g. due to a test exec failure), the successive call of the probing agent overwrites the environment information. The workflow can also deprecate existing knowledge. For example, test failures are relative to an uninstalled or unsupported package on working OS platform.

```
DIAGNOSIS:  CMake configured the tree with tests disabled (or
built in a directory that ..., so no test targets were generated;
required OS dev packages (zlib, libssl, python3) are missing in a
fresh container and must be installed before building tests ...

PRIORITIZED:  step-by-step FIXES
1) apt-get update ... \# install minimal build/test deps
2) mkdir -p build-test ...  \# force test targets
4) cmake --build ...  \# run tests (serial for clearer)
...
6) If still no tests: run ...; enable requiredlibs and reconfigure.

Documentation/files/folders to revisit
- Documentation/Building.md
- CMakeLists.txt and cmake/* (flags/options)
- test/ and test/test.sh
- build-test/CMakeCache.txt and build-test/CMakeFiles/*
```

Figure 3: Example advice from the advisor agent after examining command history and outputs. In addition to the step-by-step fixes, the agent also point out important files that are ignored and should be re-visited such as *Building.md*. This can be highly effective to guide meta agent in the re-iteration to collect essential contexts.

## 4 EVALUATION

To understand to how much aid can `USEAgentPlus` provide to human engineers in their operational tasks, we propose two research questions:

- How effective can `USEAgentPlus` be in automatically building a given project and running its test suites without human intervention? In this task, the agentic system is tasked to compose

a self-contained Bash script in a clean and controllable environment. The script alone should automaically install all the dependencies and discover testing commands.

- How effective can `USEAgentPlus` compose continuous integration (CI) configuration scripts? In this task, the agents are required to produce configuration scripts for GitLab CI/CD Platform having two stages: environment setup and test execution.

## 4.1 RQ1: CAN THE AGENTIC SYSTEM UNDERSTAND THE DEPENDENCIES OF THE ENVIRONMENT LEADING TO SUCCESSFUL TEST EXECUTIONS?

**Dataset.** We used the data set of 50 open source libraries that were used in a recent work (Bouzenia & Pradel, 2025a), covering C/C++, Java, Python, TypeScript, JavaScript, Kotlin, Assembly and Shell. Each run starts in a clean Ubuntu 24.04 Docker container environment, where the agent must generate a Bash script to build and test the project from scratch. The results presented for the `ExecutionAgent` are drawn from their publication (Bouzenia & Pradel, 2025b), specifically the configuration without web-search to allow for fair comparison between agents.

**Tool Settings.** We compare our tool against aforementioned baselines while executing `USEAgentPlus` with different settings for ablation purpose:

- OpenAI `codex-cli`, a command line coding agent that can work locally from a Bash environment that can view, edit, and run scripts in the chosen directory. We allow `codex-cli` to use any commands in a controlled environment to maximize its performance.

- `OpenHands` CodeActAgent (Wang et al., 2024c; a;b), is a commercial platform for software development agents powered by AI. we use its command line version coding agent that has similar functionality with `codex-cli`. Similarly, we allow any command to be executed during its experiments. Lightweight manual intervention provided to the tool to make it proceed with the LLM recommended option.

- *ExecutionAgent* (Bouzenia & Pradel, 2025b), a state-of-the-art academic system employs a predefined agentic workflow to generate build scripts and execute the test suite of a project source code.

- `USEAgentPlus` We evaluate our tool with re-iteration numbers set to $\{0, 1, 2, 3\}$ for ablation analysis, as our design relies on a re-iteration strategy using the advisor agent and self-reflection. When the number is set to 0, the advisor agent is disabled.

**Evaluation.** Our evaluation metric departs from exit-code metrics by examining a test summary with the numbers of Passed, Fail, and Skip (PFS) standard. Tools are forbidden from using existing CI/CD scripts or external web search, ensuring evaluation is based on the same ground.

Table 1: The results of executing test suites for 50 OSS projects. When $iter = 0$, the advisor role is disabled.

| Tool | # of successfully built | # of successful test exec. | Avg. Cost ($) |
|---|---|---|---|
| `codex-cli` | 13 | 13 | - |
| `OpenHands` CodeActAgent | 34 | 24 | - |
| *ExecutionAgent* | 31 | 24 | - |
| `USEAgentPlus` (iter=0) | 21 | 12 | 0.195 |
| `USEAgentPlus` (iter=1) | 24 (+3) | 18 (+6) | 0.453 |
| `USEAgentPlus` (iter=2) | 36 (+15) | 30(+18) | 0.498 |
| `USEAgentPlus` (iter=3) | 39 (+18) | 31(+19) | 0.683 |

The results are summarized in Table 1. Among the three systems, our tool achieved the highest performance, successfully building 39 projects and running tests on 31 of them. By comparison, *ExecutionAgent* built 31 projects with 24 test runs, while `codex-cli` achieved 13 builds and 13 test runs. In addition, the numbers by `OpenHands` CodeActAgent are 34 and 24, respectively. These results highlight the superior robustness and effectiveness of our approach.

**Ablation analysis** It is worth noting that when the re-iteration count is zero or one, both build success and test execution rates are substantially lower than those achieved by *ExecutionAgent*. When the iteration number is increased to 2 and 3, the results of build and test execution are improved significantly, outperforming all the baseline methods and this further underscores the effectiveness of the re-iteration strategy. These results also highlight the value of retrospectively examining agent trajectories. We further evaluated the monetary overhead of our approach: with three iterations, the average cost per project is only $ 0.683, suggesting it as a financially accessible tool. Fig 4 showcases a real script composed by our tool for a fresh Ubuntu container.

To determine whether environment probing contributes to the overall performance, we evaluated `USEAgentPlus` by disabling probing agent. In this configuration, the system successfully built only 30 projects, resulting in 26 successful test executions. Although this represents a clear drop compared to the full setting, the ablated version still slightly outperforms the strongest baseline method.

```bash
#!/usr/bin/env bash
set -vxE -o pipefail
# Install OS dependencies non-interactively
export DEBIAN_FRONTEND=noninteractive
apt-get update -y
apt-get install -y --no-install-recommends \
    build-essential \
    cmake \
    pkg-config \
    libssl-dev \
    ca-certificates
# Create clean build directory
rm -rf build-cmake
mkdir -p build-cmake
cd build-cmake
cmake .. -DCMAKE_BUILD_TYPE=Release
cmake --build . -- -j"$(nproc)"
ctest --output-on-failure 2>&1 | tee ../test_results.log
```

Figure 4: A generated run-test script for *libevent*. The script is able to build the project and invoke test command in a fresh Ubuntu 24.04 image container.

**SWE-Bench Verified** We revisit *SWE-Bench Verified* (Chowdhury et al., 2024), a popular repair benchmark also used for the original `USEAgent` of 500 instances and has a gold-standard test-suite used for its evaluation pipeline. However, for most practitioners, such an ideal environment does not exist. As such, we pose the agent with an empty ubuntu container including common, project agnostic tooling (grep, tree, etc.). Comparing with the official evaluation harness from *SWEBench Verified* project, we find most data points (70.2%) are able to build and run tests within their posed program repair task. The reported successful runs result from observing the last identified test-command used by the agent and check of its logs(i.e. checking that the tests run correctly and are not missing dependencies). The program repair error sources consist of overfitting (e.g. specifically adjusting to one input value), too ambitious changes (e.g. refactoring a method, failing different tests) and issues in model behavior (c.f. section 5).

## 4.2 RQ2: ADVANCING DEVOPS - CROSS-PLATFORM CONFIG GENERALIZATION TEST

To assess the extent to which the agentic system can assist in constructing CI pipelines, we conducted a cross-platform configuration generalization test. Specifically, we tasked the system with composing configuration scripts for the GitLab CI execution platform using open-source projects hosted on *GitHub*. This requires the system to rewrite its knowledge of environment setup, test execution into a structured format, namely YAML scripts. Moreover, performing cross-platform migration of CI configurations helps mitigate the risk of direct memorization of existing CI scripts, since GitLab's CI configuration logic and syntax differ significantly from those of GitHub and there are no existing GitLab scripts for these projects (the subject libraries not hosted on GitLab).

You are an experienced developer working on a Python project, with the source code located in the current directory. Your task is to write a complete and working 'GitLab CI/CD' configuration file (named 'gitlab-ci.yml') that includes environment setup, running the test suite.

...

\# Your task

Please generate the GitLab CI/CD configuration file according to the guidelines above. Try to verify the files integrity and function to the best of your capabilities.

1. Do not Introduce placeholder variables, unless they are commonly used placeholders available in a standard 'gitlab' instance

2. Do not just the above example(s)

3. Do not introduce new files to the project, except for the CI/CD File

Your final output will be a file named 'gitlab-ci.yml' ! Example structure (you can deviate from this, and from all commands presented):

```yaml
stages:
  - setup
  - test
setup_environment:
  stage: setup
  image: python:3.12
  script:  # build the project
run_tests:
  stage: test
  image: python:3.12
  script:
      # activate the environment & run test command
```

Figure 5: Task description for CI configuration (some text are skipped for better presentation).

**Evaluation Setup.** We choose 20 Python libraries that are most starred projects on Github as of June 2025 that are publicly hosted on GitHub, and AI agents are assigned to create a CI configuration script for the GitLab platform. To base all the tools on the same fresh environments, LLM agents are tasked with writing a pipeline for each subject, while they are prompted to use Python 3.12 as the default container image as shown in Fig. 5. Furthermore, the tools are allowed to use *gitlab-ci-run* for evaluation and feedback [1]. The purpose of this setting is to base all the tools on the same fresh environments. As for evaluation, we not only look at if a CI pipeline task is completed, but also check if a test summary can be found (the same metric used in as RQ1) as the test summary is the most essential outcome from a CI run.

**Baselines.** In this study, we compare our approach with `codex-cli`, `OpenHands` CodeActAgent and `USEAgent`. All tools are allowed to run any commands while taking the same task description.

The results are shown in table 2. For the environment setups, `USEAgentPlus` achieved the highest success rate (17/20), followed closely by `OpenHands` CodeActAgent (14/20), while `codex-cli` (9/20) and USE (6/20) lagged behind. As for test exections, our proposed solution still achives the best, giving 11 successful runs, outperforming `OpenHands` CodeActAgent (8/20) and `codex-cli` (4/20). The significant improvement from `USEAgentPlus` to `USEAgent` in both categories underscore the effectiveness of our design(c.f. section 3).

## 5 DISCUSSION

**Erroneous LLM Behavior.** A major source of errors observed when working with `GPT-5-mini` was the disobedience to instructions at various points, seen in figure 5. Commonly ignored were meta-level instructions such as *"there is no human in the loop"* or *"do not assume anyone else is doing the task for you"*, but even very technical instructions such as *"Do not write Code Comments"*

---

[1] https://github.com/firecow/gitlab-ci-local

| Project | Env Setup | | | | Test | | | |
|---|---|---|---|---|---|---|---|---|
| | codex | OH | USE | USEPlus | codex | OH | USE | USEPlus |
| ansible | | ✓ | | ✓ | | | | |
| django | ✓ | ✓ | | ✓ | | | | |
| flask | ✓ | ✓ | | ✓ | ✓ | ✓ | ✓ | ✓ |
| keras | ✓ | | ✓ | ✓ | | | | |
| langchain | | ✓ | | ✓ | | ✓ | | |
| matplotlib | | | ✓ | ✓ | | | | |
| numpy | ✓ | ✓ | | ✓ | | | | |
| request | ✓ | ✓ | ✓ | ✓ | | ✓ | | ✓ |
| scikit-learn | | ✓ | | ✓ | | | | ✓ |
| astropy | | ✓ | | ✓ | ✓ | ✓ | | ✓ |
| boto3 | ✓ | ✓ | ✓ | ✓ | | | | ✓ |
| distcc | | ✓ | | ✓ | | | | |
| llama_index | | | | ✓ | | | | ✓ |
| networkx | | ✓ | | ✓ | | ✓ | | ✓ |
| pandas | | ✓ | ✓ | ✓ | | ✓ | | |
| pytest | ✓ | ✓ | ✓ | | | ✓ | | ✓ |
| scipy | ✓ | | | | ✓ | | | ✓ |
| seaborn | ✓ | ✓ | | ✓ | | | ✓ | ✓ |
| tensorflow | | | | | | | ✓ | |
| scrapy | | | ✓ | ✓ | ✓ | ✓ | | ✓ |
| Total | 9/20 | 14/20 | 6/20 | 17/20 | 4/20 | 8/20 | 3/20 | 11/20 |

Table 2: The evaluation of CI scripts for 20 open sourced projects using prominent state-of-the-art agents. codex=codex-cli as of Aug 2025, OH=*openhands* as of Aug 2025, USE=`USEAgent`.

---

**Example self-identified doubts (django__django-11099)**

I could not complete an automated test run in this environment because importing Django failed due to a missing 'distutils' module (ModuleNotFoundError). I attempted to install system packages but could not provision a compatible distutils in this environment, so tests were not executed here. [...] I recommend running the test-suite in an environment where Django tests run [...].

Figure 6: Self-Identified doubts of the agent. While presented convincingly, the agent was explicitly instructed to install necessary packages, execute tests and never delegate tasks or assume other environments.

have been repeatedly discarded. `GPT-5-mini` further gave up when facing issues, like a missing dependency, instead of attempting an installation as outlined in its instructions.

Our results show that mitigation with an LLM-as-a-judge (c.f. table 2) is feasible, as such violations are observable by both humans and LLMs. However, we already employ frontier LLMs, state-of-the-art tools, and an agent-ensemble — reiteration on partly trivial instructions is a suboptimal solution. It also comes with the practical limit, as LLM judge can only perform a finite number of checks.

**Resistance to instructions and unsolicited suggestions.** We observe that LLM try to offer unsolicited suggestions or decisions similar to known *Shutdown Resistance* (Wr et al., 2025). When a command is expected to run for a long time and require significant resources, the agent may choose not to execute. Moreover, the agent tends to pose question to the user, despite it is disallowed in the system message. This tendency to refrain from executing commands and act on its own initiative could potentially disrupt software services in a production scenario, resulting in more economy loss.

> **An unsolicited decision made by the agent**
>
> ... but I did not re-execute the final altered script to capture a fresh run of only the reduced test set. If you want, I can rerun the script now (non-interactively) and attach the complete fresh logs for the final test invocation...

## 6 THREATS TO VALIDITY

**Data Leakage & Memorization**   Composing *run-test* bash scripts (c.f RQ1) and *gitlab* configuration files (c.f. RQ2) does not have a public standard groundtruth, and while it is still possible that the project's documentation memorized through the history of their public projects even there has been no explicit benchmark published yet.

**Researcher & Interpretation Bias**   It is possible that we introduce bias towards our expected findings. This paper presents our best effort of neutral judgment through two-author labelling, and we support transparency by providing logs and artifacts publicly available[2]. We aim to avoid bias in methodology, e.g. on test-time scaling, through an ablation study. Moreover, all the design and prompts are not task-specific including the advisor agent and the workflow of multiple agents in this approach is fully autonomous. Our evaluation works cover various tasks including writing Bash script after interacting with environments, repairing software bugs, and composing configuration YAML scripts while keeping the knowledge obtained from its interaction with execution environments. This illustrates our intention of avoiding overfitting datasets from the first place.

## 7 CONCLUSION

This work investigates the applicability of existing software engineering agents for continuous integration tasks, which requires substantial human effort in industry. We show that unadjusted *ReAct* agents are outperformed by specialized agents, which motivated us to implement additional, specialized members to an agent ensemble. The proposed system demonstrates robustness across a variety of tasks, improving over the state of the art in test-execution and providing CI-pipelines through infrastructure-as-code. We conducted large-scale experiments on real-world, open-source projects, and the results indicate existing approaches are versatile enough to support CI once adjusted.

**Use of LLMs in writing**. The authors have used ChatGPT for academic writing, limited to only correcting grammar issues and synonym and antonymous word suggestions.

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
