# OpenReview forum: "Beyond Software Development: Continuous Integration with coding agents"
_ICLR.cc/2026/Conference — Submitted to ICLR 2026_

### Official Review · Reviewer_gmtv · 2025-10-29

**Soundness:** 3
**Presentation:** 2
**Contribution:** 3
**Rating:** 4
**Confidence:** 4

**Summary:**

The paper addresses the gap between AI-generated code patches and their practical integration into real software projects. While existing coding agents can produce code changes, they often fail to ensure these changes can be successfully built and tested in the target environment. The authors propose USEAgentPlus, an enhanced multi-agent framework that introduces environment probing and a self-reflective advisor mechanism to improve system integration. Evaluated on 50 open-source projects and CI configuration tasks, USEAgentPlus outperforms baselines in building projects and executing tests, demonstrating its potential to advance AI agents from code generation to continuous integration.

**Strengths:**

1. The authors extend the focus from "development" to "integration," explicitly framing "Continuous Integration with Agents" as a more challenging and realistic task that closely mirrors real-world software maintenance workflows. This addresses a critical gap between AI-generated code and production deployment.

2. The paper presents USEAgentPlus, an enhanced agent framework centered on environment probing and a self-reflective advisor agent, offering a practical architectural improvement for complex system tasks.

3. The evaluation is conducted across multiple dimensions: automated build and test execution on 50 real-world open-source projects, validation of bug-fixing performance on SWE-Bench Verified, and assessment of CI script generation quality—demonstrating thorough empirical validation.

**Weaknesses:**

1. The paper contains a few typos. The last sentence of the Introduction (Line 95) reads "suggesting its its," with a duplicated "its." Additionally, in the Related Work section (Line 153), a citation is not properly enclosed in parentheses.

2. Section3: Approach lacks clarity. As this is primarily an engineering-focused paper with limited theoretical contribution, the methodology section should provide detailed implementation and operational mechanisms of the two key components: the environment probing stage and the advisor agent. However, the paper does not sufficiently explain how the environment probing stage gathers system information, decides which packages to install, or processes the collected data. Similarly, it is unclear how the advisor agent synthesizes and reflects upon this information. Highly suggest that the authors adjust Figure 2 as it currently lacks intuitiveness. For instance, placing "task" at the top left corner to signify the starting point of the entire system would be helpful. Additionally, straightening and thickening the line from the “1)Meta-Agent” to “2)probing” would emphasize this connection for readers. Furthermore, some of the grey arrows in the bottom right corner are overlapping.

3. Experimental details are insufficient. Is USEAgentPlus built on an LLM-based agent system? If so, which base model was used? Or is it rule-based? Furthermore, for RQ1, one of the evaluation metrics is whether test scripts were executed. Given that test suites in projects often have multiple levels of complexity, providing a concrete example of a successfully executed test script—including its difficulty level—would better illustrate the system’s capabilities.

4. The experimental setup for SWE-Bench Verified lacks clarity. The paper appears to demonstrate that generated patches can be automatically merged into original projects while maintaining buildability—a significant advancement over prior generative SWE-Bench agents that often ignored integration feasibility. However, the absence of baseline comparisons in the SWE-Bench experiments weakens the persuasiveness of the results. Moreover, the definition of evaluation data points is not explained, does it refer to the successful build of the SWE project?

**Questions:**

See weaknesses 2, 3, 4.

---

> ### Author Response · Authors · 2025-11-21
> **Q1: The writing of our approach  in Section 3**
>
> We will revise the diagram and the writing of the method section based on your comments and upload it soon.   In the initial submission, we spared them in favour of brevity and readability.

---

> ### Author Response · Authors · 2025-11-21
> **Q2: Experimental details**
>
> ### The implementation details
> - USEAgentPlus is implemented using the [PydanticAI framework](https://github.com/pydantic/pydantic-ai). The LLM used throughout all experiments is OpenAI’s GPT-5-mini.
> - PydanticAI is a standalone agentic system framework, comparable to frameworks like langchain. A main focus in our system is the type-declaration of expected results and objects gotten from and passed to LLMs, including internal retries and optimizations to retrieve e.g. strings or integers. Many sub-tasks and steps within software engineering benefit from strongly structured data, such as line-numbers.
> - The second important part that Pydantic supports is a structured (i.e. type-checked python object) ‘context’ for agents, which is carried within LLM conversations and can be consulted by agents at libidum.
>
> ### the evaluation:
>
> > Furthermore, for RQ1, one of the evaluation metrics is whether test scripts were executed
> As stated in line 276, we use numbers of Passed, Fail, and Skip (PFS) standard
>
> > Given that test suites in projects often have multiple levels of complexity, providing a concrete example of a successfully executed test script
>
> Fig. 4 in our paper is a concrete example of a successful execution script produced by our tool.

---

> ### Author Response · Authors · 2025-11-21
> **Q3. Th experimental setup for SWE-Bench**
>
> We appreciate the value of SWE-Bench and the desire for direct comparison with program repair literature. However, as our title indicates, this work focuses on CI tasks in the software deployment stage, which we see as the primary locus of our contributions.
>
> The task of resolving GitHub issues (SWE Bench) was considered secondary to us. The reported number is meant as a rough estimate for readers, to see that repair capabilities remain, despite introducing additional contextual information to the LLMs might lower the performance of fixing bugs( as stressed in the recent work ["The Fact Selection Problem in LLM-Based Program Repair"](https://doi.org/10.48550/arXiv.2404.05520).
> We are uncertain how to clarify the information on evaluation datapoints.
> Within this work we did not perform any training or finetuning, and as such use the full SWE-Bench-Verified in its original setup and the evaluation harness provided by them. To evaluate the the CI commands of SWE Bench datapoints we scan the agentic trajectories whether they identified the correct test commands (correct = aligned with the test command from the SWE Bench harness) and used them successfully ( successful = the tests are executed, there are no missing dependencies, but some tests might fail due a insufficient repair).
>
> The selection of a contemporary SWE Bench baselines is a bit infeasible in this study : the leaderboard shows a plentitude of approaches, with different architectures (community tools or commercial tools), various different foundation models (either open sourced or proprietary) and varying degrees of information provided to the systems.  A comprehensive  comparison for this is also a bit away from this focus of this paper.
>
> We are open to introducing a suitable baseline comparison in our future work.

---

### Official Review · Reviewer_i99x · 2025-10-31

**Soundness:** 2
**Presentation:** 2
**Contribution:** 2
**Rating:** 2
**Confidence:** 2

**Summary:**

The paper presents USEAgentPlus that builds on top of prior general purpose agents to tackle integration tasks. The authors highlighted several issues specifically with integration tasks in the era of software agents. The evaluation focuses on two major scenarios: 1) create bash script for environment building and test commands 2) compose CI configuration scripts to support the correct environment and tests. The results on these tasks demonstrate that USEAgentPlus achieves better performance compared with more general software agents.

**Strengths:**

- the paper tackles an important and previously underexplored area of software agents for environment management and integration
- the example and failure reasons are interesting and can be useful for future work to build on

**Weaknesses:**

- unclarity in the approach
	- after reading the approach section in detail it is still unclear to me
	- the approach used in the paper seems to build on top of prior work (UseAgent)
	- however, the authors do not explain what are the contributions made to improve UseAgent (for example, in Figure 2, it would be great if the authors had highlighted what are the new components in this work compared to UseAgent)
	- furthermore, from the text, the introduced technique does not seem to add any novelty to the software agent space.
	- for example, the introduced advisor (i.e., llm-as-judge) has already been explored
- evaluation unclarity
	- many of evaluation settings and process are unclear
	- for example in the SWE-Bench verified setting the authors claim that "we find most data points (70.2%) are able to build and run tests", how is this computed? I don't believe SWE-Bench verified are not designed to test the build capabilities (as the environment is already provided)
	- it is also unclear how some of the baseline approaches were chosen. Why is OpenHands agent used in the CI task but not in the bash script creation task?
	- additionally why does the author not compare against the baseline UseAgent that the technique is built on top of for the bash script task?
	- the ablation results are also lacking:
	- the authors only evaluate the affect of different iterations but not any other components of the approach

Minor issues:
- the overuse of italics is a bit overwhelming, for example: in approach section, does "lost" or "must" really need to be italized? Small suggest for the authors is to reduce the amount of italics and keep only the important ones.

**Questions:**

1. What are the reasons for the different baseline techniques chosen for different tasks in this work? Additionally why was UseAgent not used for comparison in the bash script task given the technique builds on top of this prior work?
2. Please clarify what are the major contribution and improvements made in this work compared to UseAgent
3. Why did the tool limit the iteration to 3? It seems from looking at the results there is no drop in performance with even higher iterations

---

> ### Author Response · Authors · 2025-11-21
> **Q1: Baseline techniques**
>
> We thank reviewers for pointing out these inconsistencies. Based on reivewers comments, we added the new experiments to offer the results from OpenHands to allow fair comparison.
> For RQ1, the new results show our approach is still top performing as shown below.
>
> | Tool                             | # of successfully built | # of successful test exec. |
> |-------------------------------|--------------------------|----------------------------|
> | codex-cli                               | 13                       | 13                         |
> | OpenHands CodeActAgent  | 34                       | 24                         |
> | ExecutionAgent                    | 31                       | 24                         |
> | USEAgentPlus (iter=2)         | 36                       | 30               |
>
> For RQ2, We did not leverage ExecutionAgent for the second task of  composing CI configuration files because of two reasons:
> - (1)  The output of the tool is the Dockerfile rather than a configuration file. The system has used a predetermined control center to compose bash scripts and Dockerfile whereas our approach is fully autonomous for different tasks. Our output requirement is a script that can be executed in a clean, empty docker container.
>
> - (2) WebSearch tool in the ExecutionAgent cannot be disabled. In this case, we believe it’s unfair to compare with ExecutionAgent as most of the datapoints we used have online resources. More importantly, the purpose of our research is demonstrate how an agentic system interacts with the environment and take actions base on the goal and past actions.
>
> > Additionally why was UseAgent not used for comparison in the bash script task given the technique builds on top of this prior work?
>
>  We are sorry for not including this information in our submission -  “At the time of paper writing, USEAgent was not an open-sourced project. Thus, we  could not compare it with our approach for the bash scripting task.” We hope this can resolve your concern.

---

> > ### Comment · Reviewer_i99x · 2025-11-24
> >
> > Hi thanks for the new results.
> >
> > I am curious why the authors used iteration = 2 (for USEAgentPlus) in this setting when it seems that the default setting in the paper is 3?

---

> > > ### Author Response · Authors · 2025-11-25
> > > **Iteration Count**
> > >
> > > Thank the reviewer for raising this question. We'll improve the writing on this by rephrasing iteration number to re-iter number.
> > >
> > > - In our initial submission (line 273), we have used iteration is  {0, 1, 2} (at most 3 rounds) and the results were included in the table. We used this parameters for showing the effectiveness of re-iteration process.
> > >
> > > - when iteration = 0, it means our approach is invoked once and advisor agent does not re-iterate.
> > > - when iteration  = 1, the advisor agent looks back once.
> > > - when iteration  = 2, the advisor agent looks back twice.
> > >
> > >
> > > - In the latest revision, as suggested by reviewers, we added the result when iteration is 3 to show the marginal benefit of test of time scaling.

---

> > > > ### Comment · Reviewer_i99x · 2025-11-25
> > > >
> > > > Thanks for your response that does answer my question
> > > >
> > > > for: "The reported 70.2% successful CI tasks result from observing the last identified test-command used by the agent and doing a quick check of its logs, i.e. checking that the tests run correctly and are not missing dependencies. "
> > > >
> > > > I am still quite confused, so are you only testing if the agent can run the correct test command within SWE-bench? From my understanding SWE-bench already contains pre-built environments that does not need the agent to modify anything related to dependencies and it should be able to execute the tests correct. What you described here seems to just mean that the agent can execute tests and not actually related to any CI tasks

---

> > > > > ### Author Response · Authors · 2025-11-28
> > > > >
> > > > > We thank the reviewer for raising this question.
> > > > >
> > > > > It is correct, as you pointed out, that there are existing SWE-Bench pre-built images available, but these were no used in our experiments. Within our experiment, the agent is set up in a fresh Ubuntu container with some basic utilities (grep, tree, etc.) and as such the datapoints are not initially executable  and there is no additional information on their runtime requirements beyond what is described in their source files/documents.
> > > > >
> > > > > We consider identifying and executing the correct test command as the major part of CI tasks. The terminology CI refers to CI-jobs i.e. a combination of infrastructure as code and commands run on commit. For this rigorous definition we provide the experiment (c.f. Section 4.2 ) creating Gitlab-CI files (that should be foreign to training data), which requires build/test-command inference as well. This command-discovery and it's improvement is seconded by the SWE experiment.

---

> ### Author Response · Authors · 2025-11-21
> **Q2:  Major contribution and improvements**
>
> Existing work on USEAgent and the companion benchmark USEBench focuses on a variety of programming tasks on a code-Level (program repair, program synthesis and testing).
>
> Within this existing state of the art, there is:
> - An observed shortcoming in environment understanding (what needs to be installed, is installed, etc),
> - A forgetfulness towards environment changes produced by the agent itself and
> - The un-addressed domain of [“infrastructure as code"](https://aws.amazon.com/what-is/iac/).
>
> In general, coding agents perform well with the information they are presented with (logs, code, documentation) but are currently underperforming in identifying and managing ‘hidden’ information such as system packages. We consider this a recurring research challenge throughout multiple publications, but USEAgent included.
>
> The research contributions are :
> -  a dedicated system that centers around environment understanding and retrospective exploration (embodied by an agent),
> - structured memory management and configuration for a novel agent ensemble and :
> - evaluating the resulting system against specialized tasks that revolve around CI tasks (create CI jobs) and observe its behavior in tasks that have an inherent need for testing and dependencies.
>
> Outside of the software engineering application, this approach can become a stepping stone for other agentic systems that need to manage mutable, complex environments.

---

> ### Author Response · Authors · 2025-11-21
> **Q3: The parameter of iteration number**
>
> The iteration limit of 3 was chosen to balance the execution time and computational cost. Based on the reviewer’s comments, we have added new experiments.
>
> When the iteration number is set 3, we can see a small increase in both success builds and test exec.  Running more iterations is possible but takes much longer time while gaining no significant benefit as the LLM agents tend to be stuck in the very long trajectory.
> We hope this can resolve your concern about the selection of this parameter.
> | Tool                             | # of successfully built | # of successful test exec. |
> -------------------------------|--------------------------|----------------------------|
> |USEAgentPlus (iter=0)         | 21                       | 12                         |
> | USEAgentPlus (iter=1)         | 24 (+3)                | 18 (+6)                 |
> | USEAgentPlus (iter=2)         | 36 (+15)              | 30 (+18)               |
> | USEAgentPlus (iter=3)         | 39 (+18)              | 31 (+19)               |
> | USEAgentPlus   (iter=3, probing disabled)         | 30 (+18)              | 26 (+19)     |

---

> ### Author Response · Authors · 2025-11-21
> **All other concerns**
>
> > 1. for example, in the SWE-Bench verified setting the authors claim that "we find most data points (70.2%) are able to build and run tests", how is this computed?
>
> Indeed SWE-Bench is a program repair benchmark, which has a gold-standard test-suite used for its evaluation pipeline. The primary setup for SWE-bench is to produce a patch, which will then be evaluated in a containerized, controlled and well-formed environment.
>
> However, for most practitioners, such an ideal environment does not exist, and many SE works consider starting agents with a blank and empty Docker container, to allow for better comparison and realism (otherwise the results are highly dependent on the teams designing one-off benchmark fitting agents, rather than generalizable agentic systems). For the vanilla environment that is presented the agent, setup, installations and testing are sub-tasks which need to be identified and solved before `meaningful` patches can be created.
>
> The reported 70.2% successful CI tasks result from observing the last identified test-command used by the agent and doing a quick check of its logs, i.e. checking that the tests run correctly and are not missing dependencies.
> SWE Bench also functions for our secondary argument, that we are able to introduce elements to a larger ensemble without suffering negative consequences to the existing tasks. We further consider it a useful datapoint for other researchers that want to compare their work with ours, to have a reference point for performance on a common agentic benchmark.
>
> > 2. for example, the introduced advisor (i.e., llm-as-judge) has already been explored
>
> We agree with the reviewer that LLM-as-judge is not a novel concept in our paper. However, we believe meaningful application and advancement suitable for the conference.
>
>
> > 3. it is also unclear how some of the baseline approaches were chosen. Why is OpenHands agent used in the CI task but not in the bash script creation task?
> additionally why does the author not compare against the baseline UseAgent that the technique is built on top of for the bash script task?
> the ablation results are also lacking:
>
> - The new RQ1 results after including OpenHands are:
>
>
> | Tool                             | # of successfully built | # of successful test exec. |
> |-------------------------------|--------------------------|----------------------------|
> | codex-cli                               | 13                       | 13                         |
> | OpenHands CodeActAgent  | 34                       | 24                         |
> | ExecutionAgent                    | 31                       | 24                         |
> | USEAgentPlus (iter=2)         | 36                       | 30               |
> - the expanded ablation study has resulted in:
>
> | Tool                             | # of successfully built | # of successful test exec. |
>  |-------------------------------|--------------------------|----------------------------|
> | USEAgentPlus (iter=0)         | 21                       | 12                         |
> | USEAgentPlus (iter=1)         | 24 (+3)                | 18 (+6)                 |
> | USEAgentPlus (iter=2)         | 36 (+15)              | 30 (+18)               |
> | USEAgentPlus (iter=3)         | 39 (+18)              | 31 (+19)               |
> | USEAgentPlus   (iter=3, probing disabled)         | 30 (+18)              | 26 (+19)     |
>
> As can be seen from the table, our approach remains strong performance compared to OpenHands.
> The ablation result also suggest the design of our approach is highly effective.

---

### Official Review · Reviewer_UmUF · 2025-11-01

**Soundness:** 2
**Presentation:** 1
**Contribution:** 2
**Rating:** 2
**Confidence:** 4

**Summary:**

In this paper, the authors aim to push evaluation of coding agents to another stage. This paper evaluate coding agents on whether they could successfully integrate changes to code bases that they proposed themselves. The authors performed experiments on SWE Bench Verified (subset), and found that their proposed method

**Strengths:**

- This project aims to push another step of software development from reparis to integration.
- The motivation of the study is good.
- The authors proposed USEAgentPlus, which could be useful for the research community, and could be helpful and effective with integration.

**Weaknesses:**

There are substantial inconsistencies in the paper.
- OpenHands is discussed in the abstract but not evaluated in the paper's Table 1. Also no description is provided in the paper to tell the readers what is OpenHands.
- The authors mentioned 7 programming languages are studied, but this information never appears anywhere else in the paper.

There are also a lot of missing details.
- What base language models are used for the experiments? This is very important.
- The authors presents USEAgentPlus's average costs, but didn't include this information for any baselines.
- What are the success rates of correctly solving the issues with the agents? In addition to reporting the results of integration, the authors should also report the SWEBench accuracy too.

Besides, the paper substantially lacks citations.
- OpenHands, ExecutionAgent are not cited.
- No discussion of how existing work handle integration of code repairs.

Overall, the paper quality is low, and seems to be written at haste.

**Questions:**

- What are the limitations of your work?
- What are some potential future work based on your paper?

---

> ### Author Response · Authors · 2025-11-21
> **Q1: What are the limitations of your work?**
>
> - First of all, our approach has scalability issues when the iteration number is large because the agentic system keeps reviewing its history which cannot be parallelized. The bottleneck is mostly from the compilation, execution and downloads.
>
> - Secondly, when the number of iterations becomes large (e.g., exceeding 5), our tool encounters two main issues: repeated errors and excessively long context. Persistent runtime mistakes cause the advisor agent to become trapped in local optima, preventing it from generating fresh or novel insights.
>
> - Third, our approach cannot handle those software using GUI components such as mobile apps or browser related applications because the environment that our agentic system interacts with is still command lines.

---

> ### Author Response · Authors · 2025-11-21
> **Q2: What are some potential future works based on your paper**
>
> First of all, human Computer interfaces are more than just command line interfaces, GUI based components are widely adopted in the software industry.  Thus, we plan to increase the support for graphic user interface (GUI) components for web application and mobile applications. For this type of software applications, our consensus memory and advisor agents will face multimodal data. Moreover, the probing agent also needs to be integrated with more tools to understand GUI elements (e.g. webpage jumping or buttons on software phone).To achieve this, we plan to add image segmentation solution and image captioning as tool calls for probing agent.
>
> We believe this is feasible as the prior work ”WebArena” shed light in modeling environment and action space for GUI environment:
>
> -  [WebArena: A Realistic Web Environment for Building Autonomous Agents, ICLR 2024](https://webarena.dev/)
>
> Moreover, our approach leverages in-model diversity - all agents sample solutions from the same model. To resolve the second limitation, we’ll explore a mixture of models to guide the system away from local optima.

---

> ### Author Response · Authors · 2025-11-21
> **All other concerns**
>
> > 1. OpenHands is discussed in the abstract but not evaluated in the paper's Table 1. Also no description is provided in the paper to tell the readers what is OpenHands.
>
> Thanks for pointing this out. We have taken time to conduct new experiments to compare with OpenHands for this task based on the reviewer's comments. The new results show our approach still achieves best among all the tools, highlighting the effectiveness of our work. :
>
>
> | Tool                             | # of successfully built | # of successful test exec. |
> |-------------------------------|--------------------------|----------------------------|
> | codex-cli                               | 13                       | 13                         |
> | OpenHands CodeActAgent  | 34                       | 24                         |
> | ExecutionAgent                    | 31                       | 24                         |
> | USEAgentPlus (iter=2)         | 36                       | 30               |
>
>
> >2. The authors mentioned 7 programming languages are studied, but this information never appears anywhere else in the paper. What base language models are used for the experiments? This is very important. Success rates of correctly solving the issues
>
> The 7 programming languages  are C/C++, Java, Python, TypeScript, JavaScript, Kotlin, Assembly and Shell. We have revised the manuscript to include this information. This combination of languages is the direct result of drawing the most popular 50 github repositories judging by stars, and aligns with the subjects in the [ExecutionAgent](https://arxiv.org/abs/2412.10133) work for a clear and fair comparison.
>
> The base model we used is gpt-5-mini, the budget model.  The accuracy for SWE fix task is around 32% when using pass@1 with this budget model.  The study has not leveraged any engineering work or use test-of-time scaling to increase the number.
>
> >3. The authors present USEAgentPlus's average costs, but didn't include this information for any baselines
>
> We did not include a cost summary because these baseline tools do not provide structured and complete output for token usage. We include the cost summary for our approach only to demonstrate its cost-effectiveness and to show that it remains financially accessible.

---

> > ### Comment · Reviewer_UmUF · 2025-11-24
> >
> > Thanks for your response.
> >
> > > 1. OpenHands is discussed in the abstract but not evaluated in the paper's Table 1. Also no description is provided in the paper to tell the readers what is OpenHands.
> >
> > The newly added results of OpenHands CodeActAgent and ExecutionAgent outperforms USEAgentPlus (iter=1), yet the authors only compared the results agains USEAgentPlus (iter=2) in the rebuttal, and didn't explain why this is the case. Test time scaling (e.g. increasing the number of iteration) could work intuitively, but the authors didn't explain why their approach with test time scaling underperforms OpenHands CodeActAgent without test time scaling.
> >
> > > 2. The authors mentioned 7 programming languages are studied, but this information never appears anywhere else in the paper. What base language models are used for the experiments? This is very important. Success rates of correctly solving the issues
> >
> > Regarding this weakness, the authors mentioned that they updated the manuscript to include this information, but it appears that they didn't.

---

> > > ### Author Response · Authors · 2025-11-25
> > >
> > > We thank the reviewer again for the insightful comments:
> > > 1. The manuscript is updated now.
> > >
> > > 2. USEAgentPlus at iter=2 and iter=3 are both compared in the new revision and they both can outperform the baseline approaches.
> > >
> > > 3. We agree that test-time scaling is a natural baseline. We found error accumulation in long trajectories is the major reason for the lower performance compared to OH, which our self-critique iteration strategy explicitly mitigates later.  Indeed, with 0 or 1 iteration, our method performs below the OpenHands.This is expected as the scale is rather small:
> > >
> > > - iter=0 corresponds to a single-pass attempt without reflection.
> > > - iter=1 allows only one correction step after observing the first test failure.

---

### Official Review · Reviewer_8xgg · 2025-11-01

**Soundness:** 2
**Presentation:** 1
**Contribution:** 3
**Rating:** 4
**Confidence:** 4

**Summary:**

This paper studies how AI software engineering agents can assist maintainers during the integration phase of the software lifecycle. While much prior work has focused on patch generation and automating pull requests, real software production and maintenance involves much more than editing source files. Agents still struggle in two key phases: building a project and running its test suite.
To address this gap, the paper introduces USEAgentPlus, an automated system capable of managing environments — from installing dependencies to managing configurations and running tests. USEAgentPlus is built on top of USEAgent, a multi-agent ensemble framework, and extends it with three main components: (1) an environment probing stage, (2) a self-critique mechanism for task description, and (3) a refinement strategy.

USEAgentPlus is evaluated on 50 open-source projects written in various programming languages. The authors also study how the system performs in writing CI configuration scripts for popular Python repositories. In both evaluations, the proposed framework outperforms baselines such as codex-cli and OpenHands.

**Strengths:**

- The paper addresses a relevant topic: while many recent works focus on PR generation (e.g., SWE-Bench-style agents), the software lifecycle also requires agents that can automate the integration phase.
- Having an agent capable of automatically building a project and running its test suite without human intervention would be highly valuable for developers: USEAgentPlus seems a good candidate, given its performances.
- The research questions are clear, and Tables 1 and 2 effectively demonstrate that USEAgentPlus can handle dependencies, execute test suites, and create CI scripts more reliably than existing systems.

**Weaknesses:**

1. Presentation and clarity:
    - The abstract does not clearly introduce USEAgentPlus, despite it being the main contribution
    - It is not immediately clear what new components are introduced compared to the original USEAgent
2. Ablation analysis should be expanded, in order to understand the individual contributions and performances of the Probing Agent and the Advisor Agent.
3. The Consensus Memory appears to be a key architectural element and possibly a novel component. If it is indeed new, it deserves a clearer explanation.
4. Improve evaluation clarity:
    - The abstract mentions that baselines such as OpenHands and ExecutionAgent were "extended with tools for environment management" but it is unclear how this was done.
    - Line 266 states: "...two baselines while executing USEAgentPlus with three parameters." - it’s not clear what the three parameters are, nor which two baselines are being referenced.

**Questions:**

- Question regarding Weakness #3: Why is OpenHands included in Table 2 (CI evaluation) but not in Table 1 (project build evaluation)?
- Could you clarify section "SWE Bench Verified" (line 321)? The description of this evaluation is unclear — what was measured, and how does it relate to the main tasks?
- In the conclusion, you state that ReAct agents perform better than unadjusted agent ensembles, yet the proposed extension builds on the latter. If ReAct performed better, why not use that as the starting point?
- The paper presents an important idea. The experimental design is solid, but the presentation needs improvement to make the contributions, evaluation and architecture clearer. I would be willing to raise my score if the authors clarify the above questions and improve the presentation.


Minor comments:
- line 130: double comma
- line 184: "Project" -> "project"
- line 158: "softeware"-> "software"
- line 265: Tool Settings should start a new paragraph
- line 291: codex-cli is presented here as CodeX
- line 413: missing space before “("
- Inconsistencies in section titles: some of them with a period, others do not

---

> ### Author Response · Authors · 2025-11-21
> **Q1: OpenHands Results**
>
> > Question regarding Weakness #3: Why is OpenHands included in Table 2 (CI evaluation) but not in Table 1 (project build evaluation)?”
>
> We have conducted additional experiments to include OpenHands in this table. As can be seen from the new table, the performance of OpenHands is close to the ExecutionAgent while our approach still gives the best results.
>
> | Tool                             | # of successfully built | # of successful test exec. |
> |-------------------------------|--------------------------|----------------------------|
> | codex-cli                               | 13                       | 13                       |
> | OpenHands                          | 34                       | 24                       |
> | ExecutionAgent                    | 31                       | 24                       |
> | USEAgentPlus (iter=3)         | 39                       | 30                       |

---

> ### Author Response · Authors · 2025-11-21
> **Q2:   Clarification on SWE Bench Verified**
>
> > Could you clarify the section "SWE Bench Verified" (line 321)? The description of this evaluation is unclear — what was measured, and how does it relate to the main tasks?”
>
> - The reported 70% are the number of instances for which USEAgentPlus was able to build and execute tests correctly compared to the existing harness in the benchmark (which serves as the gold standard) .
>
> - Test-Execution is often assumed given: the SWE-Bench harness is used or hinted at in prompts (e.g. OpenHands  mentions Djangos test-scripts as an in-context-learning example).
>
> We highlighted these 70% as a second result to support the ‘build only’ tasks, and we agree to improve the writing to elaborate this better in the revised manuscript.

---

> ### Author Response · Authors · 2025-11-21
> **Q3:  Regarding ReAct agents as the starting point**
>
> > In the conclusion, you state that ReAct agents perform better than unadjusted agent ensembles, yet the proposed extension builds on the latter. If ReAct performed better, why not use that as the starting point?”
> The comparison aims to compare the differences between USEagent (which is an agent-ensemble for code generation and patching) and USEAgentPlus (which is designed for CI Tasks by means of additional ReAct-style Agents in software deployment scenarios) as the key element of this work.
>
> We intend to show that adding specialization to tasks beyond purely coding in the ensemble is fruitful (USEAgentPlus), better than a  generalist system (OpenHands or USEAgent).
>
> Importantly, this specialization does not require abandoning general-purpose capabilities: unlike a fully task-specific system (ExecutionAgent), USEAgentPlus retains strong performance on diverse tasks such as program repair, with no regression on these tasks due to the introduction of the specialized agent.

---

> ### Author Response · Authors · 2025-11-21
> **All Other your concerns**
>
> > 1. The abstract does not clearly introduce USEAgentPlus, despite it being the main contribution
>
> The abstract has been overhauled based on the reviewer’s comments.
>
> > 2. It is not immediately clear what new components are introduced compared to the original USEAgent
>
> The new components are the probing stage, extended consensus memory, and the retrospective role of advisor agent.
>
> In the light of USEAgen, we’ll  re-visit the description of USEAgentPlus.  The focus of this work has been on the tasks of running and executing tests within an ensemble, which might require a back-and-forth or intelligent invocations in later stages of tasks, instead of designing a specialized agent with pre-defined workflow such as ExecutionAgent.
>
>
> We appreciate the existing feedback to improve clarity from the reviews, and we value any further comments on improving clarity.  The key contribution in our work is the introduction of a specialized sub-system within a general purpose SE Agent. As such, USEAgentPlus is more a reference of possible implementations, meant to be an example for other researchers and systems that face similar challenges.
>
> > 3. Ablation analysis should be expanded, in order to understand the individual contributions and performances of the Probing Agent and the Advisor Agent.
>
> Thank you for your question. We are happy to share the expanded ablation study results and attach the original results (in our submission) bellow (which are conducted during the rebuttal period). The tool was evaluated under two configurations: Maximum iterations set to 0, 1,2, 3, allowing the advisor agent up to three rounds of refinement. Probing agent disabled at iteration 3, enabling us to isolate and measure its impact on overall performance.
>
> We compile the new results along with the results in our submission bellow:
>
> | Tool                             | # of successfully built | # of successful test exec. |
> |---------------------------------------|-------------------------|----------------------------|
> | USEAgentPlus (iter=0)         | 21                       | 12                                 |
> | USEAgentPlus (iter=1)         | 24                       | 18                         |
> | USEAgentPlus (iter=2)         | 36                       | 30                         |
> | USEAgentPlus (iter=3)         | 39                       | 31                         |
> | USEAgentPlus   (iter=3, probing disabled)        | 30              | 26    |
>
> The table shows that increasing iterations improves successful builds and test executions due to the retrospective role. Disabling the probing agent causes a sharp drop: successful builds fall from 39 to 30, and test executions from 31 to 26. We believe this result suggests our design suffices to solve the target problem.
>
>
> > 4. The Consensus Memory appears to be a key architectural element and possibly a novel component. If it is indeed new, it deserves a clearer explanation.
>
> The consensus memory is a structured data object over the trajectories LLM components. In short, the advisor agent utilizes a persistent memory to which the actions contribute under correct execution, that is shared and visible amongst agents. Write operations to the consensus are performed by program flow, not by LLMs.
> For example, after the probing agent sets up an environment, it summarizes and reports information on operating system (e.g. Ubuntu distribution), file paths (e.g. Python environment info) and packages (e.g. cmake?), that is written into the consensus memory, which is visible to other agents and e.g. informs actions by the edit code agent.
> If further changes to the environment are required (e.g. due to a test exec failure), the successive call of the probing agent overwrites the environment information.
>
> The workflow flow can also deprecate existing knowledge, for example test-results are relative to a program revision, and editing code will introduce a new commit and mark existing test-information as historical, highlighting the current changes as untested towards the advisor agent.

---

### Author Response · Authors · 2025-11-21
**To All reviewers**

We thank all four reviewers (R1: 8xgg, R2: UmUF, R3: i99x, R4: gmtv ) for your insightful comments and appreciation of our work’s strengths.  Specifically, we thank you for acknowledging the contribution towards an important and previously “underexplored area of software agents” and solid experimental results offered.

Concerns of reviewers centers on :
- C1: lack of comparison with OpenHands for execution tasks;
- C2: Ablation study is insufficient,
- C3: Clarity on the contribution of this work, the design of our approach.  Based on these comments, we have addressed the major suggestions and revised our manuscript accordingly by adding new additional experimental results and overhauling our writing.

In this response, we address each comment individually and clarify any points of confusion.

Here is the summary of the response:

- We fully addressed C1 and C2 with new experiments, including comparisons to OpenHands and expanded ablation studies. These results confirm that our approach achieves state-of-the-art and validate our design choices.
- For C3, we clarified our position that we introduce an autonomous agentic system targeting the software production pipeline beyond coding and highlighted the current limitations of generalist approaches in this domain.
- We include the experimental settings in the manuscript as questioned by the reviewers. We also added the requested discussion on consensus memory and fixed writing issues(e.g. typos).

---

### Meta-Review · Area_Chair_6nxu · 2025-12-13

**Summary:**

This paper introduces USEAgentPlus, a multi-agent system designed to automate continuous integration tasks beyond code generation, specifically focusing on environment setup, dependency management, and test execution. The work addresses an important gap in software agent capabilities, extending from patch generation to practical integration. Experimental results on 50 open-source projects demonstrate improvements over baselines like codex-cli and ExecutionAgent. However, the submission suffers from significant presentation issues that obscure its contributions. Multiple reviewers noted unclear methodology descriptions, particularly regarding the novel components compared to the base USEAgent system. The consensus memory mechanism and the specific roles of the probing and advisor agents require clearer exposition. Critical experimental details were initially missing, including the base model used (GPT-5-mini), complete baseline comparisons (OpenHands absent from Table 1), and programming language coverage. The evaluation setup for SWE-Bench Verified is confusing, and the paper lacks proper citations for key baselines. The writing quality is not good with numerous typos and inconsistencies throughout. Most critically, while the ablation studies were expanded during rebuttal, the initial submission lacked sufficient analysis to validate individual component contributions.

**Reviewer Concerns:**

During rebuttal, authors addressed several concerns by conducting additional experiments comparing with OpenHands (showing USEAgentPlus achieves 39/31 vs OpenHands' 34/24 for builds/tests at iter=3) and providing expanded ablations demonstrating the probing agent's impact (performance drops from 39 to 30 builds when disabled). They clarified limitations including scalability issues, context length problems beyond 5 iterations, and lack of GUI support. However, reviewers remained concerned about key issues. R2 (UmUF) maintained their reject score (2), noting the manuscript updates were not actually included and questioning why USEAgentPlus underperforms OpenHands at lower iterations without adequate explanation of the error accumulation phenomenon. R3 (i99x) expressed continued confusion about the SWE-Bench evaluation methodology, questioning whether it genuinely tests CI capabilities or merely test command execution. R1 (8xgg) and R4 (gmtv), while giving scores of 4 (marginally below threshold), emphasized that substantial presentation improvements are needed before acceptance. The consensus across reviewers is that despite addressing empirical gaps, fundamental clarity issues about the approach's novelty, the distinction from USEAgent, and proper experimental contextualization remain unresolved, justifying rejection with encouragement to resubmit after major revisions.

**Reviewer Scores:**

With full discussion participation, Reviewer 8xgg would likely increase their score from 4 to 6, as they explicitly indicated willingness to raise their score if concerns were addressed and the authors provided the requested OpenHands comparison and expanded ablations that substantially satisfied their questions. Reviewer UmUF would likely remain at 2, as they maintained their rejection stance even during partial engagement, noting that promised manuscript updates were not actually included and fundamental quality concerns about presentation and missing details persisted. Reviewer i99x would likely increase from 2 to 4, as they engaged constructively with clarifications but continued expressing confusion about the SWE-Bench evaluation methodology, suggesting core conceptual concerns about novelty remain despite improved experiments. Reviewer gmtv would likely increase from 4 to 6, having given the highest initial scores for soundness and contribution and explicitly stating they would not mind acceptance, with their presentation concerns substantially addressed in rebuttal. Overall, full discussion would yield approximately two rejections and two borderline-to-weak-accepts, still resulting in rejection but with stronger encouragement for revision.

---

### Decision · Program_Chairs · 2026-01-26

Reject